# Four different mechanisms for switching cell polarity

**Filipe Tostevin**[1☯], **Manon Wigbers**[1,2☯], **Lotte Søgaard-Andersen**[3], **Ulrich Gerland**[1]*

**1** Physics of Complex Biosystems, Physics Department, Technical University of Munich, Garching, Germany, **2** Arnold Sommerfeld Center for Theoretical Physics and Center for Nanoscience, Ludwig-Maximilians-Universität München, München, Germany, **3** Max Planck Institute for Terrestrial Microbiology, Marburg, Germany

☯ These authors contributed equally to this work.
* gerland@tum.de

**Data Availability Statement:** All relevant data are within the manuscript and its Supporting information files.

**Funding:** This work was supported by the German Research Council (DFG) within the framework of

## Abstract

The mechanisms and design principles of regulatory systems establishing stable polarized protein patterns within cells are well studied. However, cells can also dynamically control their cell polarity. Here, we ask how an upstream signaling system can switch the orientation of a polarized pattern. We use a mathematical model of a core polarity system based on three proteins as the basis to study different mechanisms of signal-induced polarity switching. The analysis of this model reveals four general classes of switching mechanisms with qualitatively distinct behaviors: the transient oscillator switch, the reset switch, the prime-release switch, and the push switch. Each of these regulatory mechanisms effectively implements the function of a spatial toggle switch, however with different characteristics in their nonlinear and stochastic dynamics. We identify these characteristics and also discuss experimental signatures of each type of switching mechanism.

## Author summary

Cell polarity is key to processes such as cell growth, division, differentiation, and motility. Polarity arises from asymmetric distributions of proteins in the cell. How asymmetric patterns develop from uniform protein distributions, has been studied extensively. However, it is less clear how cells can switch such protein patterns in response to a signal. Here, we identify four qualitatively different mechanisms for how a polar protein pattern can be reversed. For each mechanism, we describe experimental signatures permitting their identification in natural systems. By providing possible regulatory circuits for these mechanisms, we also offer blueprints for synthetic implementations of switchable cell polarity, in artificial or engineered cells.

## Introduction

Cell polarity is manifested in molecular and morphological asymmetries of the cell. From bacterial to mammalian cells, cell polarity is essential in a multitude of functional contexts,

the Transregio 174 "Spatiotemporal dynamics of bacterial cells" (to L.S.-A. and U.G.) and within the framework of the Grauduate school for Quantitative Biosciences Munich (QBM) (to M.W.), by the Volkswagen Foundation (to U.G.), by the Max Planck Society (to L.S.-A.), and by the Joachim Herz Foundation (to M.W.). The funders had no role in study design, data collection and analysis, decision to publish, or preparation of the manuscript.

**Competing interests:** The authors have declared that no competing interests exist.

including cell migration, cell division and differentiation, cell-cell signaling, development and tissue homeostasis [1, 2]. One fundamental question related to cell polarity is how an initially symmetrical cell can establish a polarized state and subsequently maintain it [3]. However, cells are also known to dynamically change their polarity, e.g. reversing polarity in response to external or internal signals to control motility [4–6]. This raises a second fundamental question: Which mechanisms permit reliable switching of cell polarity?

The first question, about establishing and maintaining cell polarity, is well studied, both on the conceptual level with theoretical approaches and on the experimental level by characterizing model systems. The polarization of an initially nonpolarized cell is a symmetry breaking phenomenon: In the case of essentially isotropic cells, e.g. budding yeast or epithelial cells [3], the continuous angular symmetry is broken by polarization, whereas discrete symmetry breaking occurs for rod-shaped bacterial cells [7]. Symmetry breaking can occur spontaneously [8], but is often controlled by upstream guiding cues [9], and noise can play an important role [10, 11]. While the detailed molecular mechanisms underlying cell polarization differ between organisms, they often incorporate conserved G-protein based signaling systems that use multiple feedback interactions to generate asymmetric distributions on the cell membrane via a Turing instability [12]. A class of simple networks that can achieve cell polarization was explored in a synthetic biology study [13], which first showed computationally that all such networks feature one or more of the three minimal motifs 'positive feedback', 'mutual inhibition', or 'inhibition with positive feedback', and that combinations of these motifs generally polarize more reliably. The study also corroborated the latter finding experimentally, recapitulating the basic principles underlying the establishment and maintenance of cell polarization in engineered systems. Taken together, these and other results address many aspects of the first question raised above. By comparison, significantly less is known about the second question on the dynamical control of cell polarity.

Dynamically changing cell polarity is widely observed and studied in eukaryotic model systems such as migrating neutrophils [14] and amoebae [5], as well as melanoma cells [15]. Depending on the system, its genetic makeup, and the environment, cells display a variety of dynamical patterns. For instance, melanoma cells either randomly polarize into frequently changing directions, or reverse cell polarity in an oscillatory fashion, or they persistently maintain cell polarity [15]. The dynamical control of cell polarity involves signaling. For instance, cell polarity changes can be coupled to internal signals, as in the case of yeast, where the dynamics of cell polarity is co-regulated by the cell cycle [16]. Often, cells get a directional cue from the environment governing the direction of their response [5]. However, cells can also respond to non-directional cues. For instance, a temporally decreasing chemoattractant signal triggers reversals of cell polarity in neutrophils, even in the absence of a spatial concentration gradient [14]. Which mechanisms permit such reversals induced by a non-directional signal?

Rod-shaped bacteria display much of the eukaryotic phenomenology and serve as paradigmatic model systems. For instance, the Min system, used by *Escherichia coli* to localize the septum prior to cell division [17], constitutes a prime example of autonomous cell polarity oscillations. Its underlying molecular network, based on three Min proteins, was successfully reconstituted *in vitro* [18]. On a conceptual level, the cell polarity oscillations of the Min system are analogous to those of the melanoma cells, also with respect to the basic regulatory scheme, whereby a bistable system can be turned into an oscillator via slow negative feedback [15, 19]. For signal-induced (rather than autonomous) polarity reversal, the Mgl/Rom system of *Myxococcus xanthus* constitutes a prime example. Here the cell polarity, marked by MglA, undergoes intermittent reversals triggered by the upstream Frz signaling system [6, 20]. The cell polarity reversals are accompanied by reversals in the direction of cell motion, enabling motility patterns that are crucial for predatory behavior and fruiting body formation [21, 22].

Recently, Guzzo et al [23] identified the response regulator FrzX as a mediator of the Frz reversal signal to the Mgl system, and proposed a mechanism for how FrzX can interact with the three core polarity proteins to trigger polarity reversals. Here, we take this study as a starting point to explore the question of signal-induced polarity switching on a more general level. Rather than focusing on one particular mechanism, we aim to identify the distinct classes of switching mechanisms and their underlying working principles. We find four distinct classes of mechanisms that can occur for different signaling regimes. We demonstrate that some are sensitive to the amplitude and duration of the input signal but relatively robust to intrinsic molecular noise, while others are less sensitive to signal variability but more susceptible to noise. These and other features allow us to identify experimental signatures that can be used to discriminate between the four classes of mechanisms *in vivo*.

## Results

We consider a cell polarity defined by an asymmetric distribution of a certain 'polarity marker' *A*. The polarity marker has the regulatory role to direct the spatial localization or activity of downstream processes. For instance, MglA in *M. xanthus* is a polarity marker that localizes at one of the cell poles and activates the motility machinery to determine the direction of cell motion [20]. Similarly, Cdc42 is a polarity marker in yeast and other eukaryotic cells [24]. A module consisting of the polarity marker and other regulatory proteins has the ability to establish and maintain a polarized distribution of *A*. This module, which we refer to as the 'core polarity system', receives input from a signaling pathway via a signaling protein *X*. We stipulate that the 'full polarity system' consisting of *X* and the core system can implement the function of signal-induced polarity switching (Fig 1).

To explore mechanisms for signal-induced polarity switching, we consider a symmetric cell with a polarity marker that localizes only at its two cell poles '1' and '2' (Fig 1A), while it rapidly

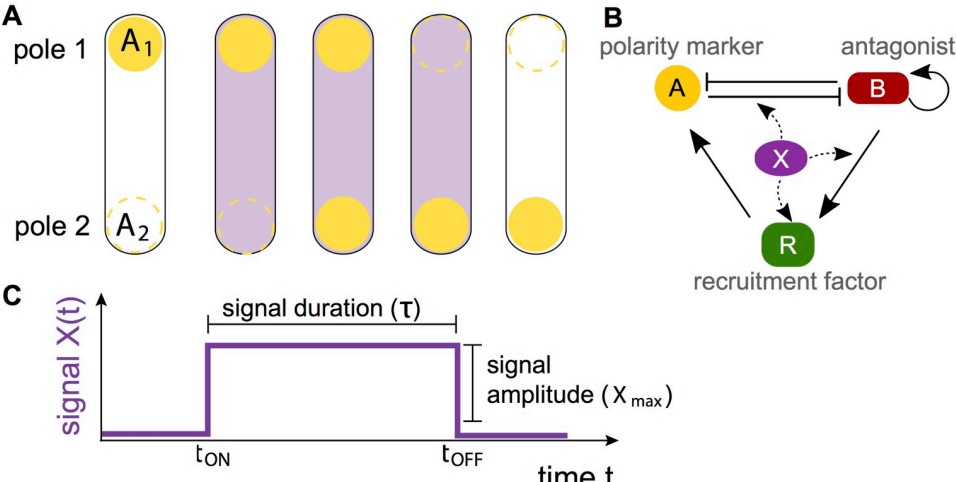

**Fig 1. Signal-induced polarity switching. A** Schematic representation of a rod-shaped cell with polarity marker *A* shown in yellow. Proteins can either be bound to the poles or diffuse in the cytoplasm. The abundances of the polarity marker at the two poles are denoted by $A_1$ and $A_2$. The release of a signal protein *X* in the cytoplasm, shown in purple, can lead to a polarity reversal, such that the polarity marker switches from pole 1 to pole 2. **B** Schematic representation of the molecular interactions of the polarity model. The polarity marker *A* and its antagonist *B* inhibit each others binding to the pole. *B* can cooperatively recruit itself to the pole and promotes binding of the recruitment factor *R*, which in turn recruits *A*. Dashed lines indicate exemplary hypothetical interactions of the signal protein *X* with the polarity proteins. **C** The switching signal is implemented as a pulse in the total amount of *X*, parameterized by the signal duration $\tau$ and signal amplitude $X_{max}$.

diffuses in the cytoplasm. This simplest scenario is a good approximation for the *M. xanthus* polarity system [23] and suffices to reveal general principles of signal-induced polarity switching, as we will see below. The distribution of *A* is then characterized by quantifying its abundance at pole '1' and '2', as well as in the cytoplasm, and the time-dependent cell polarity can be defined as

$$\omega_A(t) = \frac{A_1(t) - A_2(t)}{A_1(t) + A_2(t)} \ , \tag{1}$$

where $A_1(t)$ and $A_2(t)$ are the time-dependent abundances of *A* at the poles. Hence, $\omega_A > 0$ corresponds to a higher abundance of *A* at pole 1 than at pole 2, and vice versa for $\omega_A < 0$, such that a reversal of cell polarity is marked by a change of sign in $\omega_A(t)$.

## Model for a switchable polarity system

To obtain our working model, we generalize the recently proposed model of the *M. xanthus* polarity system [23]. This model, consisting of three compartments (pole 1, pole 2, and cytoplasm), involves the 'antagonist' *B* to the polarity marker *A*, as well as a third protein species, the 'recruitment factor' *R* (representing MglB and RomR, respectively). The network of interactions between *A*, *B*, and *R* is shown in Fig 1B. Besides the mutual inhibition between *A* and *B*, it involves self-recruitment of *B*, as well as indirect recruitment of *A* by *B* via *R*. The full dynamics of the interactions between *A*, *B*, and *R* at the poles is described by [23]

$$\begin{aligned}
\frac{dA_i}{dt} &= k_{rA}(1 - A_1 - A_2)R_i - k_a A_i - k_{ba} A_i B_i^2 \\[6pt]
\frac{dR_i}{dt} &= (1 - R_1 - R_2)(k_R + k_{bR} B_i) - k_r R_i \\[6pt]
\frac{dB_i}{dt} &= (1 - B_1 - B_2)(k_B + k_{bB} B_i) - k_b \frac{k_M}{B_i + k_M} B_i \\[4pt]
&\quad - k_{ab} A_i B_i^2 \ ,
\end{aligned} \tag{2}$$

using the same convention for *B* and *R* as for *A*, i.e. $B_i$ and $R_i$ denote the abundances at the poles ($i = 1,2$). Eq 2 assume that the total abundances of *A*, *B*, and *R* in the cell are approximately constant, at least on the relevant timescale of polarity reversals. These total values are set to one by choosing appropriate units for the abundances. The dynamics in the cytoplasm is then obtained from the dynamics of the polar abundances, e.g. the cytoplasmic abundance of *A* is $1 - A_1 - A_2$. In total, the interactions between *A*, *B*, and *R* are specified by 10 rate constants and one saturation parameter. *R* binds to the cell poles with rate $k_R$ where it locally recruits *A* with rate $k_{rA}$. *B* binds at the intrinsic rate $k_B$ to the poles, where it recruits both itself, at rate $k_{bB}$, and *R* at rate $k_{bR}$. At the same time, *A* can displace *B* from the pole and vice versa with a rate $k_{ab}$ and $k_{ba}$, respectively. All three proteins can also spontaneously unbind from the poles, with the corresponding rates $k_a$, $k_r$, and $k_b$, but the unbinding of *B* is slowed in presence of more *B* (with the saturation parameter $k_M$ determining the characteristic abundance for this feedback effect).

The positive feedback from *B* onto its own localization together with the mutual inhibition of *A* and *B* allow this model to spontaneously generate a stable asymmetry in the protein abundances at the two poles. Polarity schemes based on mutual antagonism also play a role in polarity establishment of the PAR system [25] determining the anterior-posterior axis in *C. elegans*, and the Rac-Rho system regulating front-rear polarity in mammalian cells [15]. Here, we use

Eq 2 to describe the deterministic dynamics of the core polarity system. To explore noise effects due to the relatively low copy numbers of regulatory proteins within cells, we also devised a stochastic model based on stochastic differential equations, see Eq 6 in 'Methods'. These equations take the same form as Eq 2, but with an added noise term in each equation that depends on the state of the system. The noise strength in this model is determined by an effective "copy number" parameter $N$, with $N \to \infty$ recovering the deterministic dynamics and noise strength increasing with decreasing $N$.

The signaling protein $X$ mediates a non-directional signal that interacts with the core polarity system (Fig 1B) to induce polarity switching (in *M. xanthus*, $X$ corresponds to phosphorylated FrzX [23]). We assume the total amount $X_t$ of $X$ to have a step-like pulse form (Fig 1C), parameterized by an amplitude $X_{\max}$ and duration $\tau$. While step-like pulses are a reasonable assumption, given that signals change via (rapid) protein modifications rather than (slow) changes in protein levels, we will also study the effect of more gradual changes further below. In the model of [23], cytoplasmic $X$ is recruited to the poles by the antagonist $B$ with rate $k_X$ and spontaneously unbinds with rate $k_x$, such that its polar abundances change according to

$$\frac{dX_i}{dt} = k_X(X_t - X_1 - X_2)B_i - k_x X_i \; . \tag{3}$$

In order to systematically explore the possible mechanisms by which polar $X$ may interact with the core polarity system, we allowed $X$ to regulate each one of the 11 parameters in Eq 2. We allowed for both positive and negative regulation, thus obtaining 22 different candidate models for a switchable polarity system. In each case, one parameter, denoted $k_j$, depends on $X_i$ while the others are not affected (the index $j$ in $k_j$ specifies which of the 11 parameters in Eq 2 is regulated). For a positive regulation, we have

$$k_j(X_i) = k_j(1 + X_i) \; , \tag{4}$$

and for a negative

$$k_j(X_i) = k_j(1 - X_i) \; . \tag{5}$$

Hence, a candidate signaling scenario is parameterized by (i) which parameter $k_j$ is regulated by $X$, (ii) whether the regulation is enhancing or repressive, and (iii) the amplitude and duration of the pulse, as illustrated in Fig 2A.

## Identifying functional switching scenarios

To test a candidate signaling scenario for its ability to induce polarity switching, we simulate the dynamics (both deterministic and stochastic) of the model. The output of a simulation is a set of time-dependent abundances of the four proteins $A$, $B$, $R$, and $X$ at the two poles (Fig 2B). Each simulation run has three phases. First, we simulate the polarity model, Eq 2, in the absence of signaling input ($X_t = 0$). In this condition, the system reaches a stable polarized configuration. At $t = 0$, we then switch to $X_t = X_{\max}$ for a duration $\tau$, after which the simulation is continued with $X_t$ set to zero again. We then compare the polarization of the cell at the time when the signal is initiated ($t = 0$) with a time point after the removal of the signal ($t_{\text{end}} = 30$ was chosen to allow for the system to fully relax back to a polarized steady state). The candidate signaling scenario is considered to generate a successful switch if the signs of $\omega_A(0)$ and $\omega_A(t_{\text{end}})$ were different (i.e., the initial and final polarity states were different), and unsuccessful otherwise (Fig 2C). For the stochastic dynamics, we estimated the switching probability from 100 simulation runs (Fig 2B and 2C). We repeated this procedure for each signaling scenario

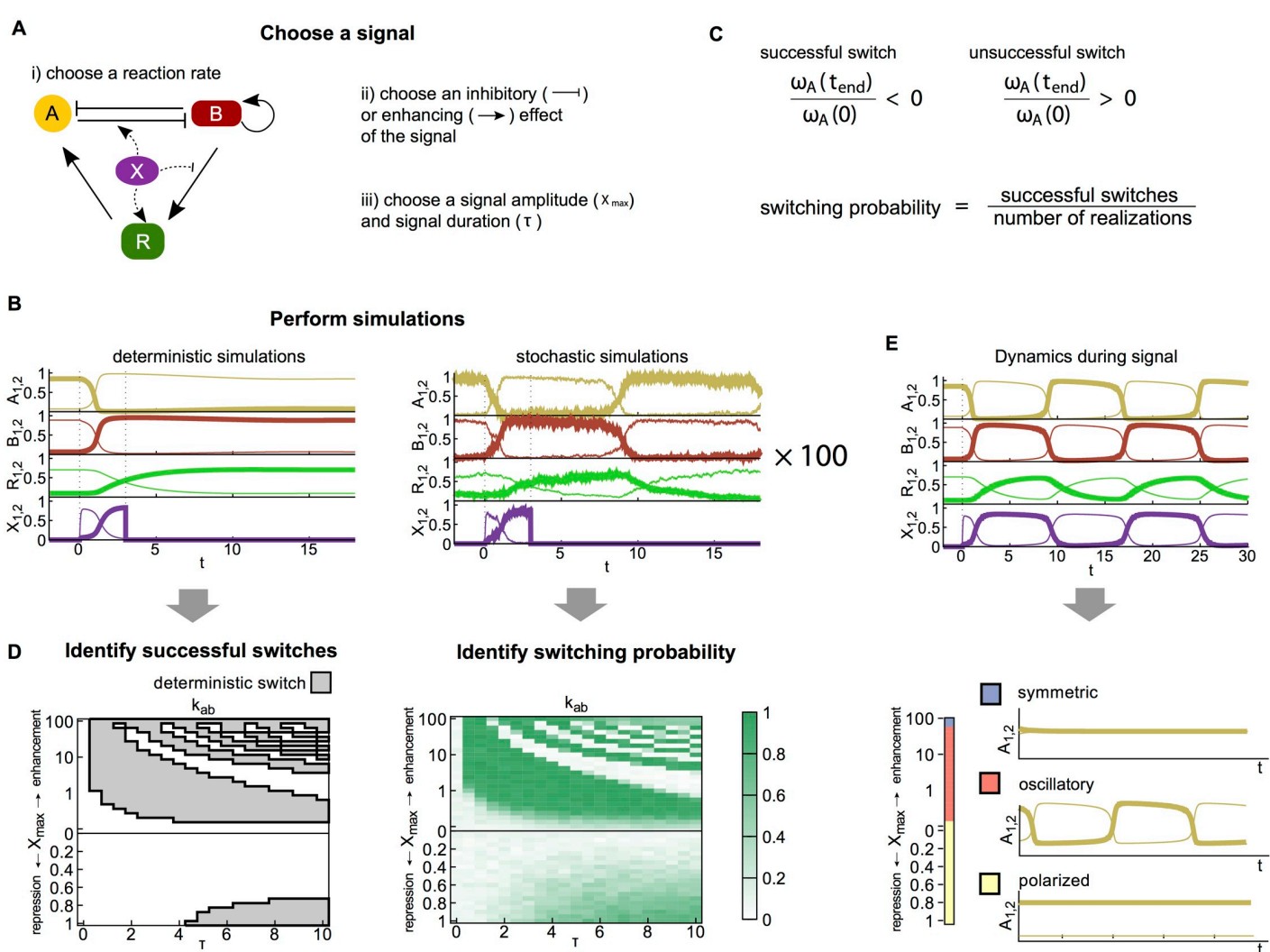

**Fig 2. Schematic representation of the workflow. A** Switching signals are parameterized by the choice of i) a reaction rate it acts on, ii) an inhibitory or enhancing effect and iii) the amplitude $X_{\max}$ and duration $\tau$ of the transient signal. $X$ can act on any of the 11 parameters of the polarity model. **B** Example of a deterministic and stochastic simulation before, during and after the signal. The signal is applied between $t = 0$ and $t = 3$. Thick lines indicate the concentrations of $A$ (yellow), $B$ (red), $R$ (green) and $X$ (purple) at pole 1, and thin lines at pole 2. **C** Switching is evaluated by comparing the signs of the asymmetry $\omega_A(t)$ in $A$ before and after the switch. For the stochastic simulation a switching probability is calculated from 100 trajectories. **D** Switching regimes are plotted in phase space as a function of $X_{\max}$ and $\tau$ for the modification of each model parameter. For the deterministic model, successful switches are shown by the gray regions with a black outline, for the stochastic model switching probabilities are shown in green. **E** The state of the system during the signal is identified by simulating the deterministic model with the signal applied for the duration of the simulations. The dynamics is classified into three states: symmetric (blue), oscillatory (orange) and polarized (yellow).

with a range of $X_{\max}$ and $\tau$ values, generating deterministic and stochastic phase diagrams delineating the functional regimes in $(\tau, X_{\max})$-space (Fig 2D).

## Characterizing functional switching scenarios

Fig 3 shows the resulting phase diagrams, each representing regulation via one of the 11 model parameters and including both enhancing and repressive regulatory effects. Here, the deterministic regimes of successful polarity reversals (solid black lines) are superimposed with the stochastic switching probabilities (green shading). We identified at least one range of signal parameters with successful polarity reversals in each of the phase diagrams. That is, it is possible for $X$ to induce reversals by regulating any of the interactions of the polarity proteins,

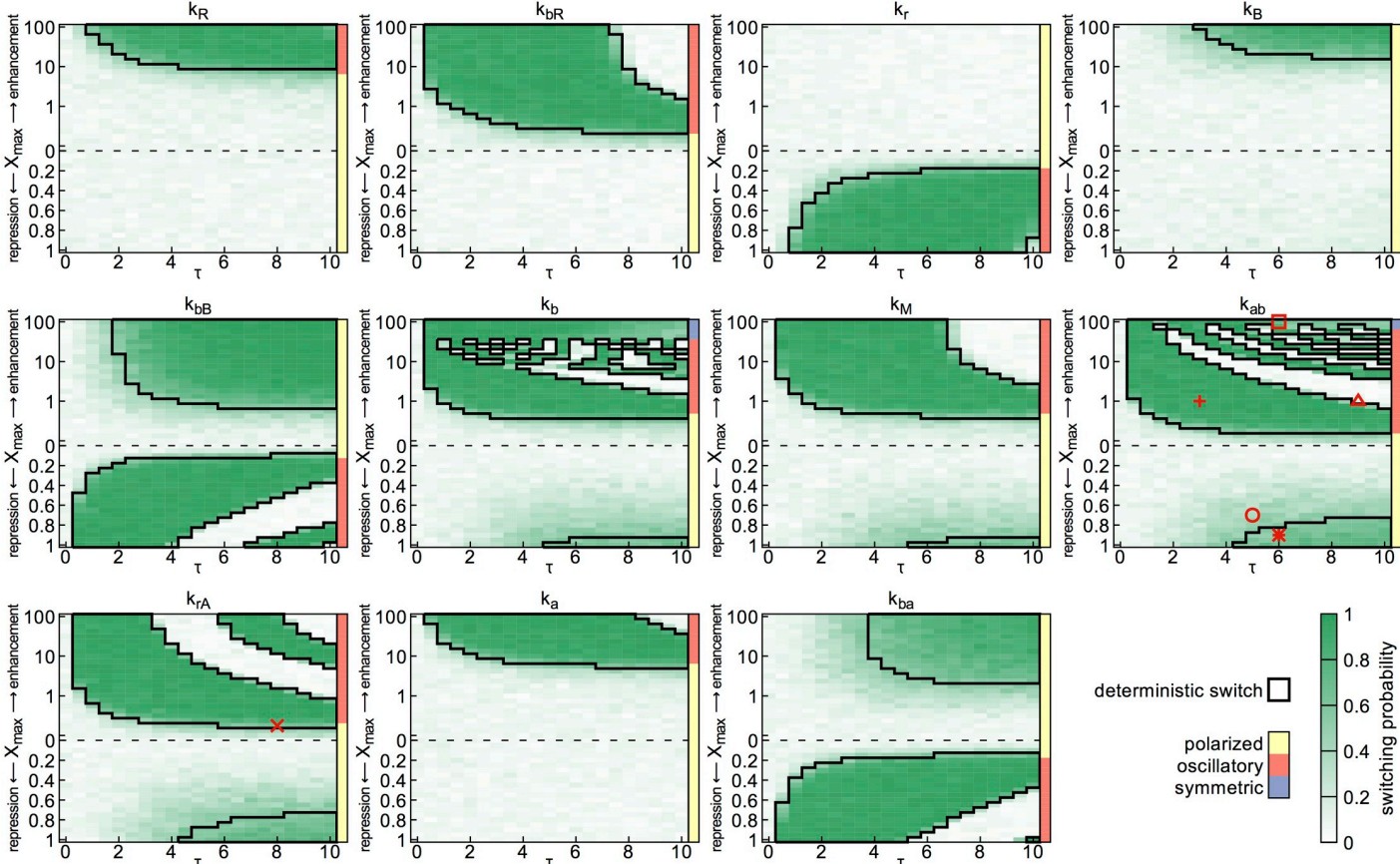

**Fig 3. Switching regimes for each of the model parameters.** Regions in which the deterministic model shows switches are indicated by thick black outlines. The green shading shows the switching probability of the stochastic model with $N = 10^{3.75}$. The upper half of the phase diagram shows results for a signal that enhances the reaction rate, and the lower half for a repression of the rate. The colored bars to the right of each panel indicate the class of dynamics when the corresponding amplitude of signal is applied, with yellow for polarized, orange for oscillatory and blue for symmetric polar distribution of $A$. The red symbols indicate the signal amplitude and duration of the trajectories shown in Fig 4.

provided that the profile of the signal pulse $X_t$ is chosen appropriately. Surprisingly, in most cases reversals can be observed when $X$ acts either positively *or* negatively. For example, reversals can be induced by $X$ either enhancing or repressing the strength of $B$ self-recruitment via the parameter $k_{bB}$.

Polarity is highly sensitive to regulation of some parameters (e.g. $k_{bB}$, $k_{ba}$), with switching occurring for most signal profiles. These parameters tend to be those involved in the key interactions of Fig 1B, including the nonlinear feedbacks in $B$ recruitment, $A$ recruitment by $R$, and $A$-$B$ mutual antagonism, which together are crucial for the establishment and maintenance of polarity. For parameters that are more peripheral to the interaction network, in particular the spontaneous binding and dissociation rates (e.g. $k_R$, $k_B$), switching occurs only in small regions of high-amplitude signals.

Fig 3 reveals two qualitatively different patterns in the signaling regimes generating switching: solid regions, in which switching is insensitive to $X_{max}$ and $\tau$ provided these exceed a threshold; and alternating bands of successful and unsuccessful switching regions, in which the system remains sensitive to the values of $X_{max}$ and $\tau$. We repeated the analysis for two other parameter sets which were randomly chosen (by multiplying each of the original parameter values $k_j$ by a random number between 0.5 and 1.5). The qualitative patterns remain as

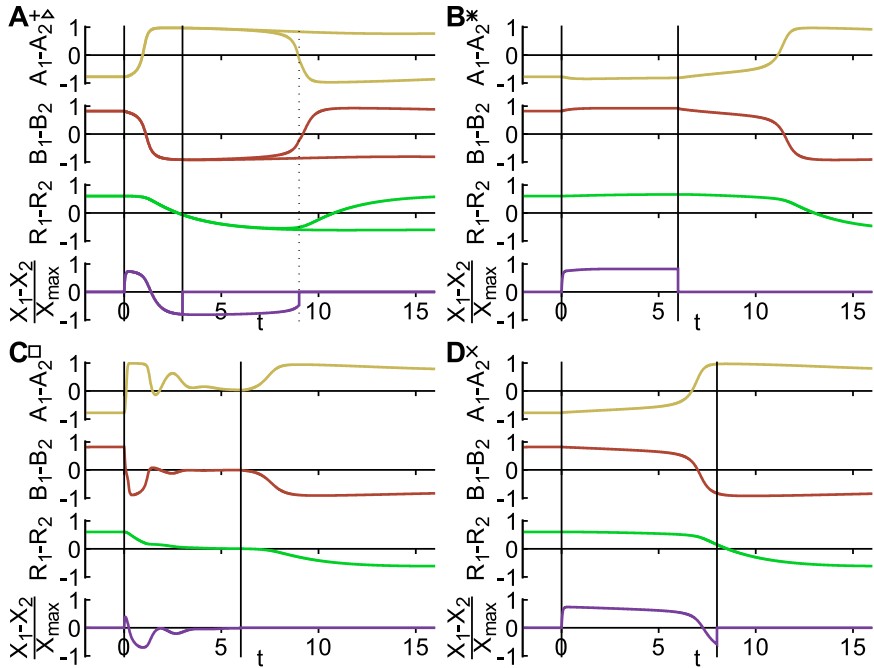

**Fig 4. Trajectories of the model during switches, classified as four different switching classes.** Signal parameters $X_{max}$ and $\tau$ and the parameter modified are indicated by the corresponding symbols in Fig 3. Vertical dashed lines indicate the period during which the signal is present. **A** Relaxation oscillator. For a short signal (plus-symbol), the polarity switches during the applied signal as shown by the solid lines. For a longer signal (open triangle), the system switches a second time as shown by the dashed lines. **B** Prime-release switch. During the signal the polarity is unchanged, but switches after the signal is released. **C** Reset switch. During the signal, the system relaxes to a symmetric distribution of the polarity marker and establishes a reversed polarity after the signal is removed. **D** Push switch. The system switches while the signal is applied and does not switch back when the signal is applied longer.

shown in S1 and S2 Figs. Intuitively, alternating bands would be expected to occur, if the system dynamics become oscillatory in presence of the signal, since Fig 3 only compares the initial and final state, such that for instance it does not discriminate between trajectories in which polarity is never reversed, and those in which polarity reverses twice.

To investigate the switching mechanism in the successful parameter regimes, we examined trajectories of the system for different signals. For a trajectory within a banded region (plus symbol in Fig 3), we see that once the signal is applied, $A$ rapidly relocates to the opposite pole, followed by $B$ and on a slower timescale $R$ (Fig 4A, solid lines). If a signal with the same amplitude $X_{max}$ is applied for a longer time (open triangle in Fig 3), a second switch takes place (Fig 4A, dashed lines). Hence the width of the bands is determined by the timescale of $R$ reorientation. This particular case, where $X$ enhances $k_{ab}$, is precisely the relaxation oscillator dynamics reported in [23].

For a trajectory in the non-band signal regime (star symbol in Fig 3), the system rapidly reaches a new steady state (with the same polarity) when the signal is applied (Fig 4B). The polarity reversal occurs after, and appears to be initiated by, the removal of the signal. To confirm that there are no longer-period oscillations during the signal period, we examined the dynamics with a signal of the same amplitude for a long duration ($\tau = 100$). The system remained stably polarized throughout this duration. Thus, this switching mechanism is qualitatively different from the relaxation oscillator reported previously. Switching is insensitive to the signal duration $\tau$, provided that it is above a threshold value. We interpret this threshold as

meaning that the signal must be present for long enough to prime the system to switch, and refer to this mechanism as a "prime-release" switch.

We then examined trajectories over the entire signal space and determined the order in which the polarity of $A$, $B$ and $R$ reversed, defined by the times at which their asymmetry $\omega$ becomes zero (S3 Fig). For almost all regimes with reversals the same order was observed (S3 Fig): first $A$, then $B$, and finally $R$. This suggests that the underlying dynamics of the trajectory between the two polarity states is similar in different switching regimes. In some limited regimes, for particularly high-amplitude signals, reversal of first $B$ and then $A$ was observed. However, these reversals were almost simultaneous. In some regimes reversals of $A$ and $B$ but not $R$ occurred. In these cases, the polarity oscillations of $A$ and $B$ were so fast that a second reversal was initiated before the much slower dynamics of $R$ could catch up to the new polarity state.

## Classification of switching mechanisms

We next examined the dynamics during persistent signals for all regulations and signal amplitudes (Fig 2E). We identified three classes of behavior (S4 Fig), reflecting qualitatively different topologies of the model's state space as shown in Fig 5. These are (i) static asymmetrically polarized protein distributions, corresponding to bistable state space with the two stable states representing the two possible orientations of polarization; (ii) oscillatory protein dynamics, corresponding to a stable limit cycle in state space; and (iii) symmetric protein distributions, corresponding to a single stable fixed point in state space. The extent of these different regimes are indicated by the colored bars adjacent to each panel in Fig 3.

This analysis confirmed that band structures in Fig 3 correspond largely to oscillatory dynamics in the presence of the $X$ signal, while solid regions correspond to regimes where the system remains bistable when the signal is applied. However, we also identified regimes presenting two additional types of switches.

For large-amplitude signals, the system can transition from an oscillatory to a monostable regime. In this scenario, while the signal is applied the system gradually relaxes towards a

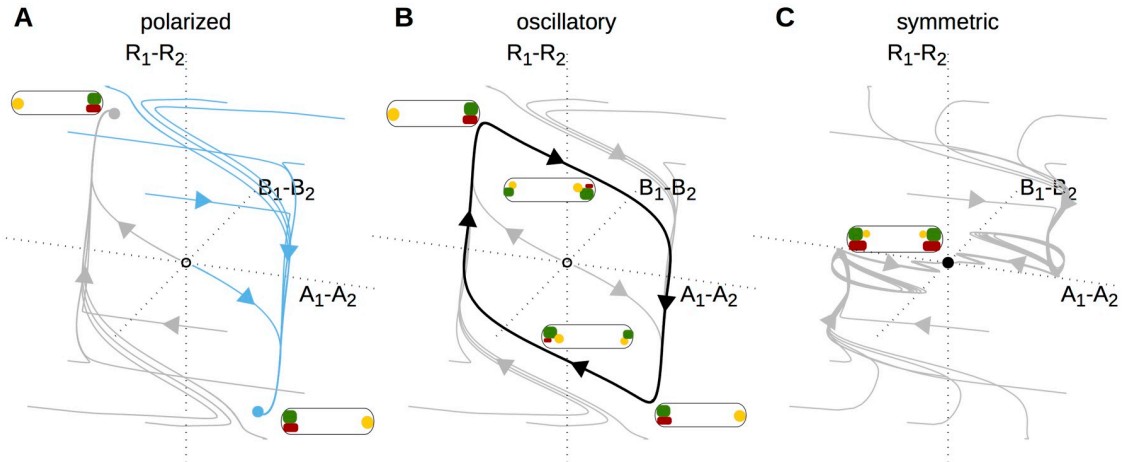

**Fig 5. In the presence of the signal, the polarity system can display three qualitatively different phase space topologies, here denoted as 'polarized', 'oscillatory', and 'symmetric'. A** For each case, the dynamics of the system is shown in the three-dimensional space ($A_1 - A_2$, $B_1 - B_2$, $R_1 - R_2$), in which the origin corresponds to a completely symmetric protein distribution. **A** In a polarized state, the system is bistable, with two stable fixed points, marked grey and blue, which correspond to the two polarities of the cell. Depending on the initial condition, the system approaches one or the other stable fixed point, as illustrated by the shown trajectories. **B** In an oscillatory state, all trajectories of the system run into a stable limit cycle, marked in black. **C** In a symmetric state, the system is monostable, with a single stable fixed point at the origin, corresponding to an unpolarized cell.

symmetric configuration (Fig 4C). Once the signal is removed, the system once again becomes polarized, but settles in the opposite polarization state from that in which it was initially. Effectively, the initial state of the system is erased and a new polarity state is chosen when the signal is removed. We therefore refer to this mechanism as a "reset" switch.

Finally, we found that as the oscillatory regime is approached, the onset of switching does not always coincide with the onset of oscillations. In the intervening region, the system still remains bistable. Examining the system trajectories, we observed qualitatively different behavior from Fig 4B. Instead of switching once the signal is removed, the system begins to switch immediately when the signal is applied, and subsequently remains stably polarized in the opposite orientation (Fig 4D). We refer to this mechanism as a "push" switch.

We have thus identified four distinct classes of switching dynamics, corresponding to four qualitatively different trajectories (Fig 4). To understand these different mechanisms from a more general nonlinear dynamics perspective, we next ask how the topology of the phase space changes in each case. Prior to the application of the signal, the system is in a bistable configuration with two stable fixed points corresponding to the two possible polarity orientations (Fig 5A). The subsequent behavior differs for each mechanism, as shown in Fig 6 and described in the following.

### Transient oscillator switch

In this class of switching, the system becomes oscillatory when the signal is applied, following the prescribed path of the limit cycle in state space. Upon removal of the signal, the phase space reverts to being bistable. The system then relaxes to one of the polarized fixed points. Which fixed point is chosen depends on the state at the end of the signal period, and in particular on which side of the separatrix (the division between the basins of attraction of the two fixed points) the state lies, as shown in Fig 6A. The duration of the signal relative to the oscillation period determines the phase at the time of signal removal and hence the final polarity state. How sensitive an oscillatory switch is to the signal duration varies dramatically between different regulations in our model, being relatively high for $k_b$ and $k_{ab}$, but low for $k_{bR}$ and $k_r$ among others.

### Reset switch

Instead of following a limit cycle during the signal period, the reset switch gradually relaxes (usually along a spiraling trajectory) towards a single stable fixed point (Fig 6B). Once again, the choice of polarity state upon removal of the signal depends only on which side of the separatrix the system is once the signal is removed. In the deterministic model, the choice of final polarity state is reliable even with a small remnant of asymmetry at the time of signal removal. However, this mechanism will be susceptible to noise in the protein dynamics that can overwhelm memory of the previous state (see below).

### Prime-release switch

This type of switch occurs when the model remains bistable even in the presence of the signal, and for parameter changes opposite to those that induce oscillations. The application of the signal does not cause a change in the topology of the state space, but does change the position of the fixed points and separatrix. If the signal is sufficiently strong, it may be that the new fixed points lie on the opposite side of the previous separatrix (Fig 6C). However, since the current state remains on the same side of the new separatrix, the system simply relaxes to the new fixed point with the same polarity orientation (the "prime" phase). Only upon removal of the

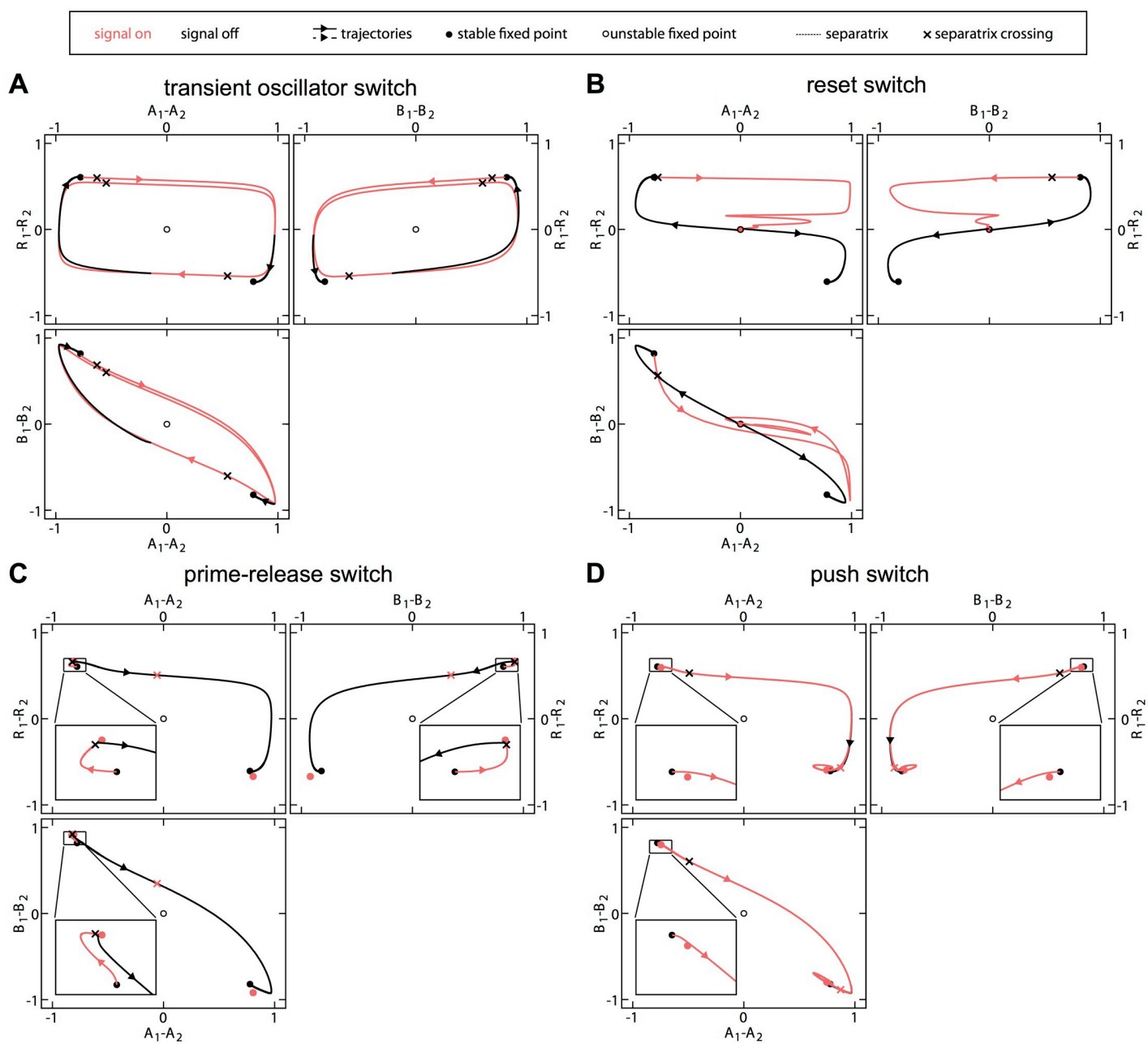

**Fig 6. Nonlinear dynamical behavior of the four different mechanisms of signal-induced polarity switching.** In each case, the system dynamics are shown both during (red) and after (black) a signal pulse, with projections onto the $(A_1 - A_2, R_1 - R_2)$-plane, the $(B_1 - B_2, R_1 - R_2)$-plane, and the $(A_1 - A_2, B_1 - B_2)$-plane. **A** Transient oscillator switch. **B** Reset switch. **C** Prime-release switch. **D** Push switch.

signal (the "release" phase) does the system find itself in the basin of attraction of the opposite polarity state.

This picture allows us to rationalize various observations about this switching mechanism. The amplitude of the signal must be sufficiently large that the new fixed point lies on the opposite side of the old separatrix, leading to a threshold in $X_{\max}$. The duration of the signal must be sufficiently long for the state of the system to move across the old separatrix, leading to a

threshold in $\tau$. Once these criteria are met, switching is insensitive to the signal amplitude and duration since the system can remain at the new polarized fixed point indefinitely.

## Push switch

The mechanism of the push switch is similar to that of the prime-release switch, but effectively with the order of events reversed. The application of the signal ("push") again leads to a shift in the positions of the bistable fixed points and separatrix, but in the opposite direction (Fig 6D). The system in its initial polarized state now finds itself on the opposite side of the new separatrix, from where it relaxes to the oppositely polarized fixed point. Upon removal of the signal, the system relaxes to the new slightly shifted fixed point but retains the same polarization. This mechanism is again largely robust to changes in the signal duration (after a threshold time needed for the initial relaxation phase), but occurs only for very small ranges of signal amplitudes in our model.

## Signals with slow edges

Both the prime-release and push switches described above rely on the fact that the signal appears and disappears very quickly, which causes a correspondingly fast change in the phase space. We expected that if the onset and removal of the signal were slower than the relaxation of the system, then the state of the system would be able to track the fixed points as they move gradually from their old to their new positions and no switching would occur. To test this prediction we computed the dynamics with the $X$ signal increasing and decreasing gradually according to $X_t(t) = X_{\max}(1 - e^{-\lambda t})$ for $0 \leq t < \tau$ and $X_t(t) = X_{\max}(1 - e^{-\lambda \tau})e^{-\lambda(t-\tau)}$ for $t \geq \tau$ (S5 Fig). We saw that for large $\lambda \gg 1$, the dynamics was similar to a step signal and switching continued to occur (S6 Fig). However for slow signals with $\lambda \lesssim 1$, switching in bistable regimes was abolished (S7 and S8 Figs). This was specific to the prime-release and push mechanisms since switching in oscillatory regimes continued to occur, with slight shifts to band boundaries reflecting the effects of the gradual signal on the oscillation phase (S9, S10, and S11 Figs).

## Stochastic effects

As seen in Fig 3, the switching probability of the stochastic model for low to intermediate noise levels tends to closely follow the boundaries of regions in which the deterministic model switches (see also S12 and S13 Figs). However, switching can also occur for signal parameters $X_{\max}$ and $\tau$ for which the deterministic system does not switch. In particular, the regimes in which switching can occur are greatly expanded by noise for prime-release and push switches, while the transition boundaries between switching and non-switching regimes of relaxation oscillators appear much sharper. For reset switches, switching remains relatively robust with short signals, which are cut off before the system has fully relaxed to a symmetric state. For longer signals the switching probability approaches 0.5, as the new polarity state is chosen randomly once the signal is removed.

Fig 7A shows the switching probability for the signal parameters indicated in Fig 3 for increasing noise level. Each mechanism displays a different noise threshold at which the switching probability departs from the deterministic result (either 0 or 1 depending on the signal parameters). This threshold is highest for the transient oscillator (+, $\triangle$), and lowest for the push ($\times$) and prime-release switches ($*$). Around $N \approx 10^{3.5}$ the switching probability converges to approximately 0.5 for all mechanisms. For higher noise levels (smaller $N$), all the mechanisms show qualitatively similar damped oscillations around 0.5. Similar behavior is observed in stochastic trajectories in the absence of any signal, indicating that these features are primarily the result of the dynamics during the period that the signal is not present ($\tau \leq t \leq t_{\mathrm{end}}$). For

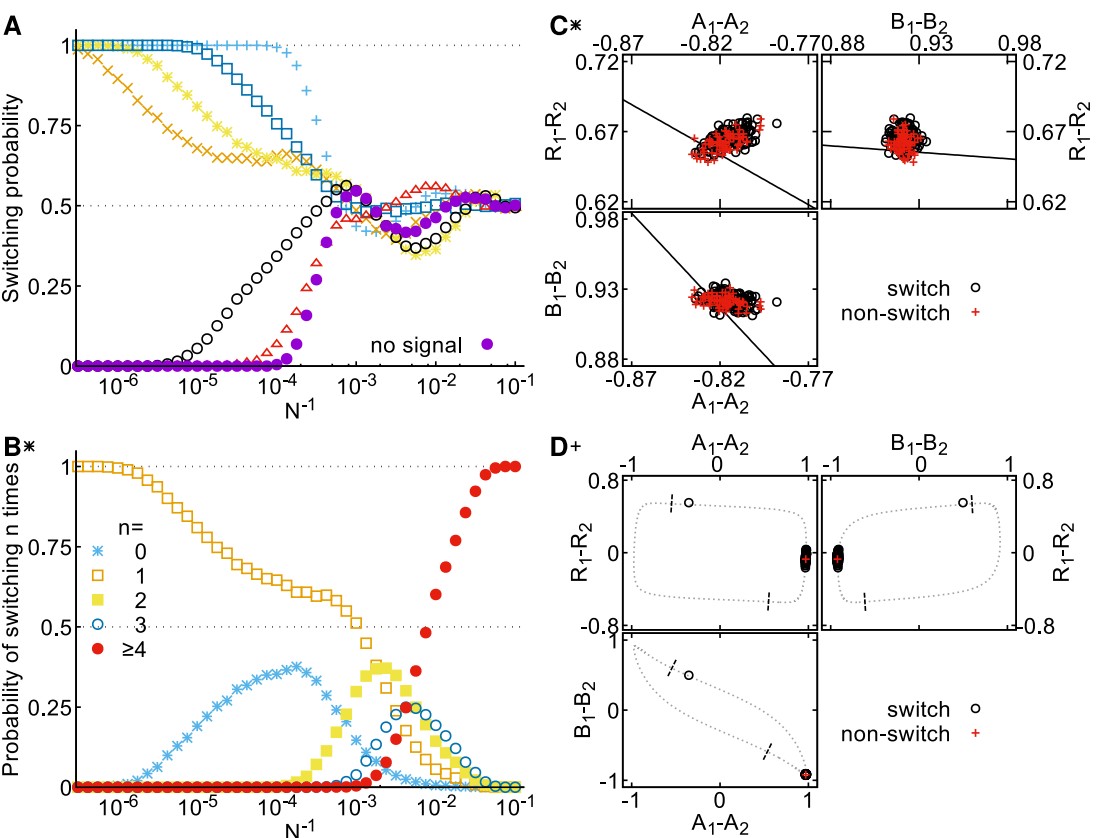

**Fig 7. Behavior of the model at different noise levels. A** Switching probability as a function of noise strength for different switching mechanisms. Signal parameters are indicated by the corresponding symbols in Figs 3 and 4. Each data point represents the results of $10^4$ stochastic realizations. **B** Probability of different numbers $n$ of switching events at different noise levels for the prime-release switch. Signal parameters are as for Fig 4B. **C** States of 200 stochastic realizations ($N = 10^4$) of the prime-release switch at $t = \tau$. Dashed line shows an estimate of the position of the separatrix in the absence of the signal (see 'Methods'). **D** States of 200 realizations of the relaxation oscillator at $t = \tau$ ($N = 10^4$). Dotted line shows the deterministic limit cycle of the system during the signal, dashed lines indicate where the limit cycle intersects with the separatrix in the absence of signal. The three different panels in **C** and **D** show different two-dimensional projections of the nine dimensional phase space. Point type and color indicate whether the system switches polarity (red) or not (black).

this reason we first focus on the regime $N \gtrsim 10^4$, in which the switching behavior remains influenced by the signal, and return to the high-noise behavior later.

## Noise-induced switching errors

The switching probability, comparing only the states of the system before and after the signal is applied, cannot distinguish between cases in which noise prevents a switch from occurring and cases in which noise causes an extra switch to occur. We therefore examined the number of polarity switching events, defined as times at which $A_1 = A_2$, in stochastic trajectories. The distributions of such events are plotted for the prime-release switch in Fig 7B (see S14 Fig for the other cases). We observe that the initial decrease in switching probability around $N \approx 10^6$ corresponds to the appearance of a sub-population of realizations that do not switch. A flattening out of the switching probability around $N = 10^4$, coincides with the appearance of trajectories exhibiting an extra second switch, due largely to stochastic switches during the period when no signal is present.

In the prime-release mechanism, switching is triggered by the removal of the signal. In the presence of noise, the system fluctuates around the fixed point of the dynamics, rather than resting exactly at the fixed point. The range of these fluctuations, visualized by sampling the states of different stochastic realizations at the end of the signal (prime) phase (Fig 7C), expands with increasing noise strength. Importantly, when the signal is removed the states of the system are clustered close to the new separatrix of the system, allowing them to be forced from the basin of one fixed point to the other by noise. The same mechanism accounts for the expansion of the range of signals for which switching can be induced in the presence of noise beyond that in which the deterministic model will switch (Figs 3 and 7A, ○). The signal is not sufficiently strong for the deterministic fixed point with the signal applied to cross the original separatrix. However, some fraction of the distribution of states around this fixed point will lie close enough to the separatrix to undergo a switch when the signal is removed. Similar behavior can also be observed for the push signal with respect to the distribution of states at the onset of the signal period.

For the transient oscillator the initial deviation from the deterministic results is due to noise-induced extra switches once the signal has been removed. Noise in the dynamics during the signal predominantly leads to phase variability, as different realizations spread out around the limit cycle. However, the state of the system at the removal of the signal is typically far from the separatrix (Fig 7D) in the slow phase of the dynamics where $R$ reacts to the new polarity of $A$ and $B$. Under these conditions, extremely high noise levels are required for the system to cross into the opposite basin of attraction. Hence, oscillatory switching appears extremely robust to noise.

## Coherence resonance

Returning to the high-noise regime ($N \lesssim 10^{3.5}$), where switching in the absence of any signal dominates, we observe that the switching probability oscillates before it saturates at 0.5 for very high noise (Fig 7A). These oscillations are reminiscent of a so-called "coherence resonance" [26]. A coherence resonance occurs when the activation timescale for noise to drive the system across the separatrix of a bistable system becomes shorter than the relaxation timescale to reach the vicinity of the opposite fixed point. The trajectory of such a stochastic system has a largely oscillatory character. Indeed, the power spectrum of the dynamics changes from monotonically decreasing at small noise to peaked at a finite frequency for larger noise (Fig 8A), indicating the appearance of oscillations. Additionally, the height of this peak shows a maximum at a finite noise level (Fig 8B), confirming the coherence resonance behavior. Thus at high noise levels, noise can drive the system between the two polarity states with a largely oscillatory dynamics, even in the absence of any $X$ signal (S15 Fig).

## Signal-induced stochastic switching

The application of a signal could also influence the stochastic switching rate during the period that the signal is active. For example, a signal could lower the height of the separatrix barrier between two fixed points, thereby increasing the chance of a stochastic switch. To study which signals could give rise to such an increase of stochastic switching, we analyzed long trajectories where signals with different amplitudes were applied continuously. Fig 8C shows the resulting mean times between switching events. We observe that indeed the mean time between switches is affected by the choice of signal. Interestingly, the mean interval between switches decreases as the signal approaches the regime of oscillations, consistent with a reduction in the height of the separatrix barrier as the bifurcation point is approached. Conversely, switches become rarer when the signal varies in the opposite direction, into the prime-release regime.

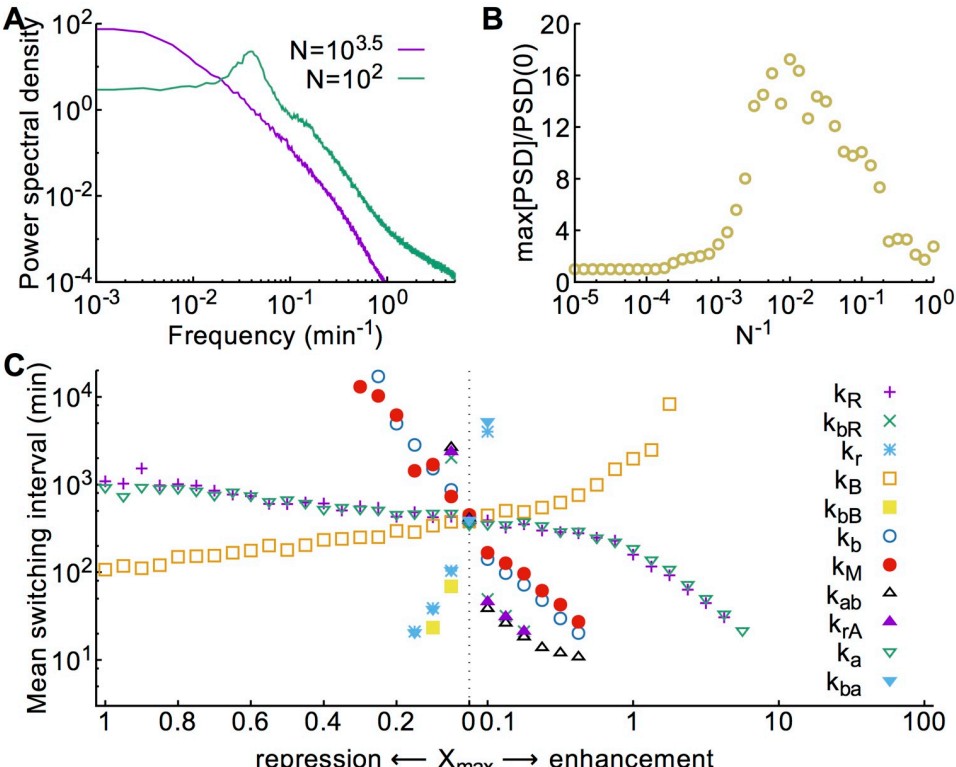

**Fig 8. Stochastic switching. A** Power spectral density of $A_1(t) - A_2(t)$ in the absence of any $X$ signal for different noise strengths. A peak in the power spectrum at high noise indicates stochastic coherence. **B** The maximal power density relative to the power at zero frequency shows a non-monotonic dependence on the noise strength. **C** The mean time between switching events, defined as points when $A_1 = A_2$, varies as different signals are applied, at a noise level $N = 10^{3.75}$. Signals that generate deterministic oscillations have been excluded. Times between switches were extracted from stochastic trajectories with the signal applied continuously for 50000 min.

In general, however, the frequency of switching is extremely low such that the expected number of switches during one signal period approaches zero. We therefore conclude that the effects of stochastic switching during the signal will be negligible and dominated by the responses of the system to the transient phases of the signal.

## Discussion

In this work we developed a classification of signal-induced polarity switching mechanisms. Our classification of switching mechanisms is not based on the molecular interactions, but on the qualitative dynamic behavior. Interestingly, one can obtain different switching mechanisms already with the same signaling and regulation network, by changing only the signal amplitude and duration, or the sign of the regulatory effect of the signal (Fig 3). Overall, we found four qualitatively different switching mechanisms: (i) the transient oscillator switch, (ii) the reset switch, (iii) the prime-release switch, and (iv) the push switch. The working principles underlying these four mechanisms can be understood already within a schematic, two-dimensional respresentation of the signal-dependent phase space structure of the system (Fig 9).

In the absence of the signal input that triggers polarity switching, the phase space structure must be that of a bistable system, with two stable fixed points corresponding to the two polarity states. The basins of attraction of these fixed points are separated by a boundary (separatrix).

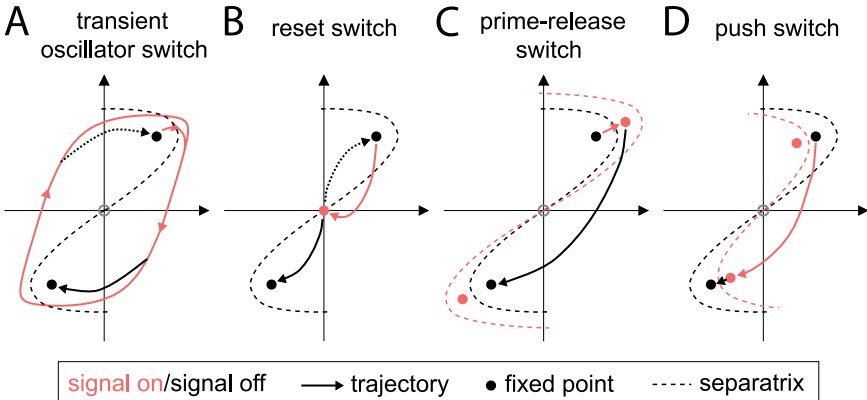

**Fig 9. Illustration of the working principles underlying the four classes of switching mechanisms. A** The different nonlinear dynamical behaviors are schematically represented in a two-dimensional phase space. Red/black symbols indicate the state space and dynamics when the signal is present/absent. Transient oscillator switch. **B** Reset switch. **C** Prime-release switch. **D** Push switch.

Before the signal is applied, the system is at one of the stable fixed points (black filled circles in Fig 9). When the signaling system is activated, it interferes with the polarity system. This temporarily deforms the structure of the phase space, and causes the state of the system to move within the phase space. The movement begins during the application of the signal (red trajectories in Fig 9), but continues after the signal has disappeared and the structure of the phase space has returned to its original state (black trajectories in Fig 9).

We found three types of phase space structure in the presence of the signal: monostable, bistable, and oscillatory (Fig 5). With these three structures, our analysis revealed four types of polarity switches. All four have in common that the temporary deformation of the phase space structure leaves the system on the other side of the separatrix when the original bistable phase space structure is restored. The transient oscillator switch achieves this by moving the system along a limit cycle during the signal (Fig 9A), while the reset switch moves it towards a single stable fixed point along a curved trajectory (Fig 9B). When the system is bistable in the presence of the signal, there are two distinct types of switches: Either the signal moves the fixed point through the original separatrix (prime-release switch, Fig 9C), or the signal pushes the separatrix through the original fixed point (push switch, Fig 9D).

The actual phase space of the system is higher-dimensional, but the qualitative behavior is the same as that shown in Fig 9. In principle, there could be polarity networks for which the signaling system induces more complex types of phase space structure, e.g. higher-order multistable or chaotic, albeit the functional benefits would be unclear. Assuming that the phase space structure is either monostable, bistable, or oscillatory in the presence of the signal, the four switching mechanisms of Fig 9 appear to exhaust the spectrum of possible behaviors. We therefore do not expect additional classes of signal-induced polarity switches to arise in other models of polarity systems with the above-mentioned properties. It is somewhat surprising that the interaction scheme of the Guzzo et al [23] model for *M. xanthus* polarity, which we took as the starting point for our analysis, is capable of producing all four types of switches. It remains to be seen whether the capacity for such diverse switching phenomenology is common to other models of prokaryotic and eukaryotic cell polarity, and which features of such models enable different switching modes. Some models, in particular those with fewer components, are likely not able to produce all four types of switches, e.g. because they cannot generate oscillations.

We also showed how the different switching mechanisms respond to signal variability and internal molecular noise. For instance, while the transient oscillator switch is most sensitive to signal variability it is least sensitive to molecular noise. By contrast, the prime-release switch is least sensitive to signal variability, but very sensitive to molecular noise. These differences in behavior will be useful as signatures to identify the actual switching mechanisms in biological systems. In addition, these properties will be relevant for the design of synthetic systems.

Currently, the *M. xanthus* system is perhaps the best studied system for polarity switching, but even there the question of the mechanism is not resolved. Guzzo et al [23] showed that the transient oscillator switch is a possible mechanism for the observed polarity switching, but other possible mechanisms are currently not excluded. Furthermore, important new components of this system continue to be found [27] and the precise interactions between the known components continue to be investigated [28]. The situation is even less clear for other experimentally studied examples of polarity switching such as neutrophils [14]. Given this state of research, it is of practical significance to know which types of mechanisms are in principle available, and what the properties of these mechanisms are.

To clearly distinguish between these mechanisms, it would be particularly useful to have experimental control over the input signal that triggers polarity switching. For the prime-release switch, the polarity reversal can only occur after the signal is removed. Hence, if the reversal is observed while the signal is still present, the prime-release switch can be excluded. The reset switch displays a loss of polarity during a long signal, which constitutes a unique signature of this mechanism. The transient oscillator switch is best detected by varying the duration of the signal. Finally, the push switch should switch only once during a long signal. Note, however, that such experiments will give insight only into the type of switch and not into the detailed interaction between the signaling protein and the polarity proteins, since the same qualitative dynamics can be generated by different modes of action of the signal. Our analysis of the systems' dynamics also revealed that, while the timing with respect to the input signal is different for the different mechanisms, the order in which the proteins of the core polarity system switch poles is almost always the same. This indicates that we cannot infer the interaction of the signaling protein $X$ from looking at the order in which the polarity proteins switch poles, but that the order of switching is rather a characteristic of the interactions between different polarity proteins.

By analogy with the paradigmatic genetic toggle switch [29], the functionality analyzed here can be regarded as a 'spatial toggle switch'. The core of the genetic toggle switch is a circuit of two mutually repressing genes, conceptually similar to the mutual inhibition between the polarity marker $A$ and its antagonist $B$. Some of the behavior is also analogous, e.g. molecular noise can cause the genetic toggle switch to flip spontaneously [30], just as it does for the polarity system. However, while the genetic toggle switch is a well-mixed bistable system, the core polarity system is a spatially extended bistable system that forms asymmetric patterns. The spatial extension of the polarity system allows a global signal ($X_t$) to be converted into a local signal (differential activity of $X$ at the two poles), in a way that would be impossible in a well-mixed system. This permits the polarity system to function as a true toggle switch, i.e. the same signal causes switching in both directions, in contrast to the original genetic toggle switch, where different signals "set" and "reset" the switch [29]. The true toggle (or "push-on push-off") functionality in genetic switches requires more elaborate regulatory circuitry that manipulates the bistable system as a function of input signals to achieve control of the system [31–34].

In comparison with genetic systems, the control of pattern forming systems is only beginning to be explored, opening interesting directions for future research. Here, we used a

simplified treatment for the pattern formation process, with the cell divided into only three regions, the two poles and the cytoplasm. The underlying assumption is time-scale separation between the diffusive transport and the relevant biochemical processes. Given typical cell lengths, e.g. $L \sim 6\mu m$ for *M. xanthus*, and diffusion coefficients $D \sim 10\mu m^2/s$ for small cytoplasmic proteins [35], the mixing timescale $L^2/(2D)$ over which cytoplasmic proteins explore the bulk of the cell is less than 2 seconds. In contrast, the observed timescale of the actual switching process, during which the abundances of the polarity system proteins decrease at one pole and increase at the opposite pole, is on the order of 30 seconds for *M. xanthus* [23], suggesting that the assumption is reasonable. However, it will be interesting to explore the dynamics also under conditions where this assumption does not hold, using full spatial models.

## Methods

### Deterministic dynamics

Reaction rates $k_j$ were chosen as in [23], with the rate $k_{ab} = 15$ min$^{-1}$. The deterministic dynamics was computed with Mathematica (Wolfram Research Inc.) using the function NDSolve separately in each domain (before, during and after the signal), with initial conditions set according to the protein abundances at the end of the previous segment.

### Stochastic model

For the stochastic version of the model we used a Langevin extension of Eq 2, adding a noise term to each equation,

$$\frac{dA_i}{dt} = k_{rA}(1 - A_1 - A_2)R_i - k_a A_i - k_{ba}A_i B_i^2 + f_{A,i}(\mathbf{x})^{1/2}\eta_{A,i}(t)$$

$$\frac{dR_i}{dt} = (1 - R_1 - R_2)(k_R + k_{bR}B_i) - k_r R_i + f_{R,i}(\mathbf{x})^{1/2}\eta_{R,i}(t)$$

$$\frac{dB_i}{dt} = (1 - B_1 - B_2)(k_B + k_{bB}B_i) - k_b \frac{k_M}{B_i + k_M}B_i$$

$$- k_{ab}A_i B_i^2 + f_{B,i}(\mathbf{x})^{1/2}\eta_{B,i}(t)$$

$$\frac{dX_i}{dt} = k_X(X_t - X_1 - X_2)B_i - k_x X_i + f_{X,i}(\mathbf{x})^{1/2}\eta_{X,i}(t)$$

(6)

where $\mathbf{x} = (A_1, A_2, R_1, \ldots, X_2)$ is the state vector, and the $\eta_{.,i}$ are independent Gaussian random variables, $\langle \eta_{.,i}(t) \rangle = 0$ and $\langle \eta_{p,\,i}(t)\eta_{q,\,j}(t') \rangle = N^{-1}\delta_{p,\,q}\,\delta_{i,\,j}\,\delta(t-t')$. We have introduced $N$ as a parameter to tune the magnitude of the noise, with the deterministic model being recovered as $N \to \infty$. We chose to make the noise multiplicative by having the strengths $f_{.,i}(\mathbf{x})$ depend on

the current state of the system, $\mathbf{x}$. Specifically,

$$
\begin{aligned}
f_{A,i}(\mathbf{x}) &= k_{rA}(1 - A_1 - A_2)R_i + k_a A_i + k_{ba} A_i B_i^2 \\[6pt]
f_{R,i}(\mathbf{x}) &= (1 - R_1 - R_2)(k_R + k_{bR} B_i) + k_r R_i \\[6pt]
f_{B,i}(\mathbf{x}) &= (1 - B_1 - B_2)(k_B + k_{bB} B_i) + k_b \frac{k_M}{B_i + k_M} B_i \\[6pt]
&\quad + k_{ab} A_i B_i^2 \\[6pt]
f_{X,i}(\mathbf{x}) &= k_X(X_t - X_1 - X_2)B_i + k_x X_i
\end{aligned}
\tag{7}
$$

We emphasize here that these noise terms were chosen simply as one plausible generalization of Eq 2. While they resemble those that might be obtained from a system-size expansion of a full Master equation for the reactions underlying Eq 2 [36, 37], we note that since the original model is defined only in terms of the rate equations and not in terms of the underlying molecular reactions, no such systematic derivation of the noise is possible. We verified that the particular choice of the form of the noise did not affect our conclusions, and found qualitatively similar results when white noise was used (implemented by fixing $\mathbf{x} = (1/3, 1/3, \ldots, 1/3, X_t/3, X_t/3)$ in Eq 7), see S16 and S17 Figs.

Simulations of the stochastic model were performed by directly integrating Eq 6 using an update rule of the form

$$
\mathbf{x}(t + dt) = \mathbf{x}(t) + dt \; \mathbf{d}(\mathbf{x}) + \sqrt{\frac{dt}{N} \mathbf{f}(\mathbf{x})} \; \xi,
\tag{8}
$$

where $\mathbf{d}(\mathbf{x})$ represents the deterministic part of Eq 6, $\mathbf{f}(\mathbf{x}) = (f_{A,1}, f_{A,2}, f_{R,1}, \ldots, f_{X,2})$ is a vector of noise strengths, $\xi$ is a vector of independent samples from a normal distribution, and multiplication of $\mathbf{f}^{1/2}$ and $\xi$ is performed elementwise. A time step $dt = 10^{-4}$ min was used throughout. After each update step, all protein abundances were corrected such that none were negative or exceeded the total protein numbers (i.e. ensuring $A_1 + A_2 \leq 1$, and similarly for each other protein). The simulation code (implemented in C++) is available at github.com/gerland-group/langevin_switching.

## Estimation of separatrices

The separatrix lines in Fig 7C,D were estimated as follows. For the prime-release switch (Fig 7C), we first estimated the state space around the fixed point in the presence of the signal by simulating 10000 stochastic trajectories with $N = 10^3$ until $t = \tau$. For each of these points, we determined on which side of the separatrix they fell in the absence of signal, by taking these as the initial conditions for deterministic simulations over the period $\tau \leq t < t_{\text{end}}$. The projections of the separatrix in the planes shown were then estimated by using a linear discriminant classifier to determine, for each of the two-dimensional projections of the data in turn, the decision boundary between the sets of states that belong to each of the basins of attraction. This analysis was performed using the 'LinearDiscriminantAnalysis' class from scikit-learn [38] with default parameters. For the relaxation oscillator (Fig 7D), we identified the path of the limit cycle from the trajectory of the deterministic model. The intersection points with the separatrix were then estimated by initializing simulations with the signal removed at different points along the limit cycle.

## Power spectra

Power spectral densities were estimated from trajectories sampled every 0.01 min for 50000 min by Welch's method of averaged periodograms from overlapping segments of the trajectory [39] using the MATLAB (Mathworks) function pwelch with segments of length $2^{16}$ samples.

## Supporting information

**S1 Fig. Switching regimes, with the signal acting on each of the model parameters as in Fig 8 of the main text, but with a different basal parameter set that was randomly chosen (by multiplying each of the original basal parameter values by a random number between 0.5 and 1.5).** In the shown example, these values were $k_{rA} = 400 \cdot 1.01$, $k_a = 2 \cdot 1.38$, $k_{ba} = 400 \cdot 0.95$, $k_B = 2 \cdot 1.15$, $k_{bB} = 30 \cdot 0.54$, $k_b = 2.8 \cdot 0.96$, $k_M = 0.3 \cdot 1.36$, $k_{ab} = 0.5 \cdot 30 \cdot 0.59$, $k_R = 0.1 \cdot 1.49$, $k_{bR} = 1.5 \cdot 0.75$, $k_r = 0.4 \cdot 0.98$, $k_X = 20 \cdot 1.16$, and $k_x = 3 \cdot 0.61$. Here, the deterministic switching regimes shift only slightly in the space of signal amplitude and duration, but the sensitivity to noise becomes significantly stronger. However, the qualitative behavior remains the same as in Fig 8 of the main text, with alternating bands and solid regions that show robust deterministic switching as long as the signal amplitude and duration exceed a threshold.
(PDF)

**S2 Fig. Switching regimes, with the signal acting on each of the model parameters as in Fig 8 of the main text, but with a different basal parameter set that was randomly chosen (by multiplying each of the original basal parameter values by a random number between 0.5 and 1.5).** In the shown example, these values were $k_{rA} = 400 \cdot 1.1$, $k_a = 2 \cdot 0.58$, $k_{ba} = 400 \cdot 1.38$, $k_B = 2 \cdot 0.81$, $k_{bB} = 30 \cdot 1.31$, $k_b = 2.8 \cdot 0.69$, $k_M = 0.3 \cdot 1.02$, $k_{ab} = 0.5 \cdot 30 \cdot 1.04$, $k_R = 0.1 \cdot 1.37$, $k_{bR} = 1.5 \cdot 1.05$, $k_r = 0.4 \cdot 1.46$, $k_X = 20 \cdot 1.08$, and $k_x = 3 \cdot 1.35$. Here, the deterministic switching regimes shift significantly in the space of signal amplitude and duration, and the sensitivity to noise becomes significantly weaker. However, the qualitative behavior remains the same as in Fig 8 of the main text, with alternating bands and solid regions that show robust deterministic switching as long as the signal amplitude and duration exceed a threshold.
(PDF)

**S3 Fig. Order of switching.** Switching trajectories are obtained from the deterministic model. Black solid lines in the phase diagrams show switching regimes as in Fig 8. The colors indicate in which order $A$, $B$ and $R$ switch polarity. In the regimes where the system switches polarity multiple times (due to the transient oscillator switch), the switching order represents the order of the first switch.
(PDF)

**S4 Fig.** Trajectories of the system during the signal for **A** the transient oscillator switch, **B** the prime-release switch, **C** the Reset switch and **D** the push switch. The symbols next to the panel labels indicate the signal parameter $X_{\max}$ as indicated in Fig 8. The signal is applied for the duration of the simulation. During the transient oscillator switch (**A**) the polarity of the system oscillates, while during the reset switch (**C**) there is no polarity, i.e. the distribution of the proteins at pole 1 and pole 2 is symmetric. During the prime-release and push switch (**B** and **D**) the system is polarized during the switch.
(PDF)

**S5 Fig. Example of a gradually increasing and decreasing signal.** The total amount of $X$, $X_t$, increases according to $X_t(t) = X_{\max}(1 - e^{-\lambda t})$ for $0 < t < \tau$ and decreases according to

$X_t(t) = X_{\max}(1 - e^{-\lambda\tau})e^{-\lambda(t - \tau)}$ for $t > \tau$. The dashed line indicates the step-like signal.
(PDF)

**S6 Fig. Trajectories for a gradually increasing and decreasing signal with λ = 4.** Signal amplitude $X_{\max}$ and duration $\tau$ are chosen the same as in Fig 4, where in **A** the solid line corresponds to the short signal (plus-symbol) and the dashed line to the long signal (open triangle). The system shows qualitatively the same behavior as for the step-like signal.
(PDF)

**S7 Fig. Trajectories for a gradually increasing and decreasing signal with λ = 2.** Signal amplitude $X_{\max}$ and duration $\tau$ are chosen the same as in Fig 4, where in **A** the solid line corresponds to the short signal (plus-symbol) and the dashed line to the long signal (open triangle). For these gradual signals, the transient oscillator switch (**A**), the reset switch (**C**) and the push switch (**D**) switch qualitatively the same as for a step-like signal, while the prime-release switch (**B**) does not respond to the gradual signal.
(PDF)

**S8 Fig. Trajectories for a gradually increasing and decreasing signal with λ = 1.** For these gradual signals, the prime-release (**B**) and push switch (**D**) do not respond to the signal, while the transient oscillator (**A**) and reset switch (**C**) do.
(PDF)

**S9 Fig. Switching regimes for a gradually increasing and decreasing signal with λ = 4.** Regions in which the deterministic model shows switches are indicated by thick black outlines. The green shading shows the switching probability of the stochastic model with $N = 10^{3.75}$. The upper half of the phase diagram shows results for a signal that enhances the reaction rate, and the lower half for a repression of the rate. The colored bars to the right of each panel indicate the class of dynamics when the corresponding amplitude of signal is applied, with yellow for polarized, orange for oscillatory and blue for symmetric polar distribution of $A$, for a gradually increasing and decreasing signal. The switching regimes are similar to the regimes for a step-like signal as shown in Fig 8.
(PDF)

**S10 Fig. Switching regimes for each of the model parameters for a gradually increasing and decreasing signal with λ = 2.** The regimes where the prime-release switch acts to switch the polarity, for example via repression of the parameter $k_{ab}$ or $k_{rA}$, have become smaller.
(PDF)

**S11 Fig. Switching regimes for each of the model parameters for a gradually increasing and decreasing signal with λ = 1.** The regimes where the prime-release switch acts to switch the polarity becomes smaller, for example by enhancing $k_B$, or completely vanishes, for example via repression of the parameters $k_{ab}$ or $k_{rA}$. In addition, the regimes where the push switch acts vanishes, for example via a slight enhancement of the parameter $k_{rA}$.
(PDF)

**S12 Fig. Switching regimes for each of the model parameters with a step-like increasing and decreasing signal.** The green shading shows the switching probability of the stochastic model with $N = 10^{3.5}$. The stochastic switching probability, outside of the deterministic switching regimes (solid black lines), is higher as compared to a noise level of $N = 10^{3.75}$ as shown in Fig 8, while the switching probability in the deterministic regimes is smaller.
(PDF)

**S13 Fig. Switching regimes for each of the model parameters with a step-like increasing and decreasing signal.** The green shading shows the switching probability of the stochastic model with $N = 10^4$. The stochastic switching probability, outside of the deterministic switching regimes (solid black lines), is smaller as compared to a noise level of $N = 10^{3.75}$ as shown in Fig 8, while the switching probability in the deterministic regimes is higher.
(PDF)

**S14 Fig. Probability of different numbers of switching for different noise levels.** Symbols next to the panel labels **A-E** correspond to the signal amplitude and duration as shown in Fig 8. **F** shown the probability of different numbers of switches without a signal.
(PDF)

**S15 Fig. Polarity switching of the stochastic model without a signal. A** for low noise levels ($N = 10^{3.5}$) the system does not switch for the duration of the simulation. **B** for high noise levels ($N = 10^2$) the polarity switches several times without applying a signal.
(PDF)

**S16 Fig. Switching regimes for each of the model parameters with a step-like increasing and decreasing signal.** The green shading shows the switching probability of the stochastic model with white noise and with $N = 10^4$. The switching regimes are qualitatively similar to the switching regimes in Fig 3.
(PDF)

**S17 Fig. Switching probability of the stochastic model with white noise.** The signal parameters are indicated by the corresponding symbols in Figs 8 and 4. Results are qualitatively similar to the results presented in Fig 6A.
(PDF)

## Acknowledgments

We thank Tam Mignot and Sean Murray for helpful discussions.

## Author Contributions

**Conceptualization:** Filipe Tostevin, Manon Wigbers, Lotte Søgaard-Andersen, Ulrich Gerland.

**Data curation:** Filipe Tostevin, Manon Wigbers.

**Formal analysis:** Filipe Tostevin, Manon Wigbers.

**Funding acquisition:** Lotte Søgaard-Andersen, Ulrich Gerland.

**Investigation:** Filipe Tostevin, Manon Wigbers.

**Methodology:** Filipe Tostevin, Manon Wigbers, Ulrich Gerland.

**Project administration:** Ulrich Gerland.

**Resources:** Ulrich Gerland.

**Software:** Filipe Tostevin, Manon Wigbers.

**Supervision:** Lotte Søgaard-Andersen, Ulrich Gerland.

**Validation:** Filipe Tostevin, Manon Wigbers.

**Visualization:** Filipe Tostevin, Manon Wigbers, Ulrich Gerland.

**Writing – original draft:** Filipe Tostevin, Manon Wigbers, Ulrich Gerland.

**Writing – review & editing:** Filipe Tostevin, Manon Wigbers, Lotte Søgaard-Andersen, Ulrich Gerland.

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
