## [Decision Letter · Decision Letter 0]

6 Aug 2020

Dear Dr. Gerland,

Thank you very much for submitting your manuscript "Four different mechanisms for switching cell polarity" for consideration at PLOS Computational Biology.

As with all papers reviewed by the journal, your manuscript was reviewed by members of the editorial board and by several independent reviewers. In light of the reviews (below this email), we would like to invite the resubmission of a significantly-revised version that takes into account the reviewers' comments.

We cannot make any decision about publication until we have seen the revised manuscript and your response to the reviewers' comments. Your revised manuscript is also likely to be sent to reviewers for further evaluation.

Sincerely,

Qing Nie

Associate Editor

PLOS Computational Biology

Jason Haugh

Deputy Editor

PLOS Computational Biology

Reviewer's Responses to Questions

**Comments to the Authors:**

Reviewer #1: The review is uploaded as an attachment

Reviewer #2: In the paper “Four different mechanisms for switching cell polarity” by Tostevin, Wigbers, Sogaard-Andersen and Gerland, the authors start with a minimal model of cell polarity oscillations, and then systematically explore the effects of different pulse inputs into the model. Through exhaustive simulations they uncover four classes of polarity switching behaviors. They characterize each switching mechanism including the effects of stochasticity by showing the switching regimes, and state space behaviors. Introducing stochastic effects increases the probability of switching including noise-driven oscillatory switching (coherence resonance).

It is surprising that the model was able to generate four distinctly different dynamical behaviors based on how the input was introduced into the model. The modeling approach was clever, introducing the input at each parameter with systematic changes in signal amplitude and length. The biological relevance of these different behaviors is less clear.

I believe that the authors have achieved a solid understanding of the novel system behaviors, but that the characterization needs more support in certain areas for a more complete story. Overall, there are interesting innovations in both approach and results.

The authors conclude that they have “developed a general classification of signal-induced polarity switching mechanisms based only on how a signaling protein changes the topology of the polarity system’s state space”. I quibble with the generality of the findings, but their treatment is broader than currently exists.

My recommendation is for a revise and resubmit with certain sections strengthened to maximize impact. My primary recommendations are listed below; more minor edits mainly focused on presentation will be provided in the next review cycle.

1. Figure 5 which is a schematic figure in the middle of the Results section needs to be replaced by phase plane diagrams generated from actual data along the lines of what is done in Figure 6D.

2. Exploration of parameter space. A key question is whether these findings are valid only for one particular parameter set (one point in parameter space) or apply broadly to a wide range of parameter values in the model. The authors need to explore parameter space, not necessarily in an exhaustive fashion but in a targeted fashion and compare the results from different parameter points in an additional figure (which can be placed in the supplement). This would address the critique whether the results are robust to parameter variation.

3. What is the biological significance of the work? In other words why is this paper in PLoS Computational Biology and not a SIAM or a Physics journal? I would like the authors to focus on two areas in the Discussion. First what are the implications regarding the possible function of FrzX. The original model has X affecting the k_{ab} term. Are some of the other simulations consistent with the data in the original paper suggesting other alternative roles for FrzX? Second, are there examples of the various spatial switching dynamics in other biological pattern forming systems? If not, the authors should say clearly that their results mainly apply to the M. xanthus system (at least for now). These points can be addressed in the Discussion.

4. What is the effect of diffusion? Again this point can be addressed in the Discussion (as a possible future direction). The model makes a simplification (which must be noted in the main text) that the interior of the cell is well-mixed. Would the results change if a time delay via diffusion is introduced as the species traverse from one pole to the other?

5. The following claim in the Discussion is not supported by any argument: “From a nonlinear dynamics perspective, the four types of behavior in Fig. 5B-E appear to exhaust the spectrum of possible behaviors, such that we do not expect additional classes to appear in other polarity models.” This sentence should be removed or supported by data.

6. I have a small number of minor edits regarding presentation. I will include these in the next review cycle.

**Have all data underlying the figures and results presented in the manuscript been provided?**

Reviewer #1: Yes

Reviewer #2: Yes

PLOS authors have the option to publish the peer review history of their article (what does this mean?). If published, this will include your full peer review and any attached files.

Reviewer #1: No

Reviewer #2: No
---

## [Decision Letter · Decision Letter 1]

1 Dec 2020

Dear Dr. Gerland,

We are pleased to inform you that your manuscript 'Four different mechanisms for switching cell polarity' has been provisionally accepted for publication in PLOS Computational Biology.

IMPORTANT: The editorial review process is now complete. Reviewer 2 has a couple of minor comments/suggestions that could be addressed in the final manuscript. Requests for major changes, or any which affect the scientific understanding of your work, will cause delays to the publication date of your manuscript.

Best regards,

Qing Nie

Associate Editor

PLOS Computational Biology

Jason Haugh

Deputy Editor

PLOS Computational Biology

Reviewer's Responses to Questions

**Comments to the Authors:**

Reviewer #1: The authors have provided a suitable response for all my comments.

Reviewer #2: The response by the authors satisfies my concerns and I recommend the manuscript for publication. I only have a few minor requests:

1) In the bottom of page 9, the authors address my point 2 (exploration of parameter space) with the line: “These qualitative patterns remain when the values of the basal parameters k_j are varied (Figs. S1 and S2). I would prefer the more complete statement that they make in their response, i.e. something to the effect: “We repeated the analysis for two other parameter sets which were randomly chosen (by multiplying each of the original parameter values by a random number between 0.5 and 1.5). The qualitative patterns remain as shown in Figs. S1 and S2.”

2) I am satisfied with the final paragraph in the Discussion in which they mention the importance of following up their interesting work with full spatial models. However, I just want it to be clearer up front that the model is a 3-compartment model (even though it should be obvious from the equations). In the first paragraph of the Results section “Model for a switchable polarity system”, I would add the following to the second sentence in this section: “This model, consisting of three compartments (pole 1, pole 2, and cytoplasm), involves the ‘antagonist’ ... “

3) The clarification of the claim of “exhausting the spectrum of behaviors” in the Discussion is good and makes sense to me.

4) My PDF had strange symbols (e.g. asterisks, crosses, etc.) next to the letters (A, B, C, etc.) in the Figures. I assume that this is a problem with my PDF.

**Have all data underlying the figures and results presented in the manuscript been provided?**

Reviewer #1: None

Reviewer #2: Yes

PLOS authors have the option to publish the peer review history of their article (what does this mean?). If published, this will include your full peer review and any attached files.

Reviewer #1: No

Reviewer #2: No

---

## [Editor Report · Acceptance letter]

11 Jan 2021

PCOMPBIOL-D-20-01142R1 

Four different mechanisms for switching cell polarity

Dear Dr Gerland,

I am pleased to inform you that your manuscript has been formally accepted for publication in PLOS Computational Biology. Your manuscript is now with our production department and you will be notified of the publication date in due course.

With kind regards,

Livia Horvath
